New whaitsioids (Therapsida: Therocephalia) from the Teekloof Formation of South Africa and therocephalian diversity during the end-Guadalupian extinction

Huttenlocker Adam K. huttenlo@usc.edu ahuttenlocker@gmail.com 1
Smith Roger M.H. 2 3
1 Department of Integrative Anatomical Sciences, University of Southern California , Los Angeles , CA , United States of America
2 University of the Witwatersrand, Evolutionary Studies Institute , Johannesburg , South Africa
3 Iziko South African Museum , Cape Town , South Africa
Marsicano Claudia
Electronic publication date: 2017 Oct 5
Publication date: 2017
Volume: 5
Electronic Location ID: e3868
Received 2017 Jul 25; Accepted 2017 Sep 8
Copyright: ©2017 Huttenlocker and Smith
Copyright year: 2017
Copyright holder: Huttenlocker and Smith
License: This is an open access article distributed under the terms of the Creative Commons Attribution License, which permits unrestricted use, distribution, reproduction and adaptation in any medium and for any purpose provided that it is properly attributed. For attribution, the original author(s), title, publication source (PeerJ) and either DOI or URL of the article must be cited.
License URL: https://creativecommons.org/licenses/by/4.0/

Keywords: Synapsid, Permian, Extinction, Tetrapod, Therocephalians

Funding: NSF DEB-1209018 DBI-1309040 Adam K. Huttenlocker has been supported by NSF DEB-1209018 and DBI-1309040. The funders had no role in study design, data collection and analysis, decision to publish, or preparation of the manuscript.

==============================
Two new species of therocephalian therapsids are described from the upper Permian Teekloof Formation of the Karoo Basin, South Africa. They include two specimens of a whaitsiid, Microwhaitsia mendrezi gen. et sp. nov., and a single, small whaitsioid Ophidostoma tatarinovi gen. et sp. nov., which preserves a combination of primitive and apomorphic features. A phylogenetic analysis of 56 therapsid taxa and 136 craniodental and postcranial characters places the new taxa within the monophyletic sister group of baurioids—Whaitsioidea—with Microwhaitsia as a basal whaitsiid and Ophidostoma as an aberrant whaitsioid just outside the hofmeyriid+whaitsiid subclade. The new records support that whaitsioids were diverse during the early-late Permian (Wuchiapingian) and that the dichotomy between whaitsiid-line and baurioid-line eutherocephalians was established early on. The oldest Gondwanan whaitsiid Microwhaitsia and additional records from the lower strata of the Teekloof Formation suggest that whaitsioids had diversified by the early Wuchiapingian and no later than Pristerognathus Assemblage Zone times. Prior extinction estimates based on species counts are reflected in an analysis of origination/extinction rates, which imply increasing faunal turnover from Guadalupian to Lopingian (late Permian) times. The new records support a growing body of evidence that some key Lopingian synapsid clades originated near or prior to the Guadalupian-Lopingian boundary ca. 260–259 million years ago, but only radiated following the end-Guadalupian extinction of dinocephalians and basal therocephalian predators (long-fuse model). Ongoing collecting in older portions of the Teekloof Formation (e.g., Pristerognathus Assemblage Zone) will shed further light on early eutherocephalians during this murky but critical time in their evolutionary diversification.

Introduction

Therocephalians were a major clade of non-mammalian therapsids whose fossils are best represented in rocks of the middle to late Permian (ca. 272–251.9 million years ago), with a few lineages that survived into the subsequent Triassic Period (Abdala, Rubidge & Van den Heever, 2008; Huttenlocker, Sidor & Smith, 2011). They attained remarkable ecological diversity with wide-ranging body sizes and myriad dental (and, thus, dietary) specializations and other modifications of their feeding apparatus (Kemp, 1972a; Kemp, 1972b; Huttenlocker, 2014; Huttenlocker & Abdala, 2015; Huttenlocker, Sidor & Angielczyk, 2015). Moreover, they are the sister-group to cynodonts—the therapsid group that includes mammals–making them exceptionally important in our understanding of the origins of mammal-like behavior and physiology (Hopson & Barghusen, 1986; Hopson, 1991; Van den Heever, 1994; Huttenlocker, 2009). Thus, there has been increasing interest in therocephalian paleobiology in recent years to clarify their evolutionary relationships (Huttenlocker, 2009; Huttenlocker, Sidor & Smith, 2011; Kemp, 2012) and to understand their fluctuating diversity patterns within the context of the global biodiversity crises that disrupted terrestrial ecosystems of the late Paleozoic—the end-Guadalupian and Permo-Triassic mass extinctions (Huttenlocker, Sidor & Smith, 2011; Huttenlocker, 2014).

Despite improvements in our understanding of therocephalian evolution, very little is known about the transition between the middle-to-late Permian, a critical period in the ecological expansion of therocephalians. Early therocephalian fossil localities are globally widespread—particularly in present-day southern Africa and Russia—but their fossils are best known from the middle Permian terrestrial vertebrate assemblages of the Karoo Basin, South Africa (Abdala, Rubidge & Van den Heever, 2008; Smith, Rubidge & Van der Walt, 2012). Basal therocephalians of South Africa’s middle Permian (Guadalupian) Tapinocephalus and Pristerognathus assemblage zones (AZ) were large-bodied apex predators that fell into only a handful of genera and species (Abdala, Rubidge & Van den Heever, 2008; Abdala et al., 2014b). By contrast, the later eutherocephalians that derived from this stock became highly diverse with up to 70 genera (Abdala, Rubidge & Van den Heever, 2008; Huttenlocker, 2013), and abruptly replaced the earlier, archaic groups in the Karoo Basin by Tropidostoma AZ times. These eutherocephalians, along with the gorgonopsians, remained among the most abundant terrestrial predators in early–late Permian terrestrial assemblages in southern Africa (Smith & Botha-Brink, 2011; Smith, Rubidge & Van der Walt, 2012). Moreover, a major dichotomy between baurioid-line and whaitsiid-line eutherocephalians may have already taken place prior to the early–late Permian Tropidostoma AZ, as some museum records of baurioid-line ‘ictidosuchids’ and whaitsiid-line hofmeyriids were collected from Pristerognathus AZ-equivalent rocks of the lower Teekloof Formation (reviewed in Huttenlocker, 2013 and this study).

Among Permian eutherocephalians, whaitsiids have received much taxonomic interest primarily due to the unconventional hypothesis that they may share a relationship with Cynodontia, thus positioning cynodonts within Therocephalia rather than as a close sister-group (Kemp, 1972a; Abdala, 2007; Botha, Abdala & Smith, 2007). Although this view has fallen out of favor (Van den Heever, 1994; Kemp, 2012; Abdala et al., 2014b), little remains known about the diversity of whaitsiid-line therocephalians or the detailed anatomy of their hypothetical common ancestor. The taxonomic history of whaitsioids is complex. Tatarinov (1974) was the first author to include his ‘Annatherapsididae’ (= Akidnognathidae), ‘Moschowhaitsiidae,’ and Whaitsiidae within the more inclusive clade ‘Whaitsioidea.’ However, Mendrez (1974) separated from Whaitsiidae the akidnognathids and other forms that she did not see fit in either of the two groups, such as the hofmeyriids Hofmeyria and Mirotenthes—previously regarded as whaitsiids or “forerunners” of whaitsiids by Watson & Romer (1956: 70) and other authors (e.g., Attridge, 1956; Brink, 1956). Early cladistic investigations of therapsids also doubted the whaitsiid affinities of akidnognathids and some hofmeyriids (Hopson & Barghusen, 1986; Abdala, 2007), although more recent revisions of therocephalian systematics and phylogeny have supported the traditional view that hofmeyriids likely do share a relationship with whaitsiids (Huttenlocker, 2009; Huttenlocker, Sidor & Smith, 2011). Consequently, two major clades of post-akidnognathid eutherocephalians are currently recognized: (1) Baurioidea and (2) a clade of whaitsiids and hofmeyriids to which the term ‘whaitsioids’ has recently been repurposed (Huttenlocker & Abdala, 2015; Huttenlocker & Sidor, 2016; Maisch, 2017) though it has not been formally defined until now.

Present study—Here, we describe two new therocephalians from the western Karoo Basin that shed light on the poorly known non-mammalian therapsid subclade Whaitsioidea. These whaitsioid records were reported in prior studies (Sidor et al., 2013; Huttenlocker, 2014; Huttenlocker, Sidor & Angielczyk, 2015; Huttenlocker & Sidor, 2016), but their anatomy is described and illustrated here for the first time. The specimens were discovered in 1996 and 2011 on two different farms near the border of the Northern and Western Cape provinces, respectively, both from outcrops of the upper Permian Teekloof Formation (Fig. 1). Teekloof-equivalent rocks of the Middleton and lower Balfour formations to the east of the study area have yielded radiometric dates that suggest a relatively continuous sequence of middle to upper Permian rocks in this part of the basin (Rubidge et al., 2013). The combination of constraints on their early–late Permian ages (Wuchiapingian) coupled with their phylogenetic position near the whaitsiid stem makes these new records an important datum for understanding the post-Guadalupian radiation of eutherocephalians and of early non-mammalian therapsid faunas more generally.

Figure 1 Specimen provenance and stratigraphic context of Teekloof Formation whaitsioids.

1, Provenance of SAM-PK-K10990 and K10984, Badshoek farm (De Hoop 117), near Beaufort West, Western Cape Province; 2, Provenance of SAM-PK-K8516, Good Luck farm, near Fraserburg, Northern Cape Province. Bulleted numbers to right of stratigraphic column indicate ages of vertebrate assemblage zones in millions of years (Ma) (from Rubidge et al., 2013). Geologic map modified from Smith (1993). Abbreviations: CiAZ, Cistecephalus Assemblage Zone; DaptoAZ, Daptocephalus Assemblage Zone; Fm, Formation; M, Member; PristAZ, Pristerognathus Assemblage Zone; TapinoAZ, Tapinocephalus Assemblage Zone; TrAZ, Tropidostoma Assemblage Zone.

Geological Context

The Beaufort Group is the most extensively exposed stratigraphic unit of the Karoo foreland basin sequence and consequently the most tetrapod fossils have been recovered from this interval. In the southwestern sub-basin, Beaufort Group sedimentation was initiated by source area tectonism and resulted in the deposition of an approximately 2,700 m thick succession of fluvial channel sandstones and overbank mudrocks (Abrahamskraal and Teekloof formations; Fig. 1) containing rich fossil tetrapod assemblages (i.e., Eodicynodon, Tapinocephalus, Pristerognathus, Tropidostoma, Cistecephalus, and Daptocephalus AZs of Fig. 1). Sandstone-rich fining-upward packages from 50 to 300 m thick, described as megacycles, occur within the succession. These packages are thought to be related to northeasterly, northwesterly, and east-southeasterly directed fluvial transport systems and subsidence-controlled shifts in the loci of channelization on the alluvial plain (Cole, 1992). The presence of calc-alkaline volcaniclastic detritus and cherts of tuffaceous origin suggests that the provenance rocks in the southwest may have included an active andesitic volcanic chain located on the eastern side of the Andean Cordillera in South America and West Antarctica. Rubidge et al. (2013) dated a series of these tuffs confirming that the Guadalupian-Lopingian boundary occurs close to the top of the Abrahamskraal Formation (Fig. 1).

The overlying Wuchiapingian-aged Teekloof Formation in which the new therocephalians were found was deposited by overbank flooding from meandering rivers of variable sinuosity draining an extensive alluvial plain sloping gently towards the northeast in the direction of the receding Ecca shoreline (Turner, 1978). Deposition occurred under semi-arid climatic conditions as evidenced by the presence of desiccation cracks, playa lake evaporite deposits with desert-rose gypsum aggregates and pedogenic carbonate nodules and lenses (Smith, 1990; Smith, 1993). The irregular accretion topography and preferential preservation of upper flow regime plane beds and lower flow regime ripple cross-lamination within channel sand bodies (Smith, 1987) indicates a flood-dominated discharge regime and seasonal inundation of the floodplains resulting in rapid bone burial (Smith, 1993).

The two new therocephalian skulls (SAM-PK-K10984, SAM-PK-K10990) collected from Badshoek (De Hoop 17) are from the same mudrock interval in the middle of the Hoedemaker Member (Tropidostoma AZ, ±257 Mya). They were found as isolated skulls and were encrusted with a 2 mm-thick layer of pedogenically-precipitated micrite. Associated dicynodont taxa collected in the same strata include Tropidostoma, Emydops, Pristerodon and Diictodon, as well as a small gorgonopsian (see Table 1).

The Good Luck (Matjiesfontein 412) specimens (SAM-PK-K8516, SAM-PK-K8631) were collected from massive gray siltstone beds in the upper Oukloof Member (upper Cistecephalus AZ, ±255 Mya) along with several skulls and partial skeletons of Cistecephalus, Oudenodon, Diictodon, Procynosuchus and Pareiasaurus (see Table 1).

Material and Methods

The electronic version of this article in Portable Document Format (PDF) will represent a published work according to the International Commission on Zoological Nomenclature (ICZN), and hence the new names contained in the electronic version are effectively published under that Code from the electronic edition alone. This published work and the nomenclatural acts it contains have been registered in ZooBank, the online registration system for the ICZN. The ZooBank LSIDs (Life Science Identifiers) can be resolved and the associated information viewed through any standard web browser by appending the LSID to the prefix http://zoobank.org/. The LSID for this publication is: urn:lsid:zoobank.org:pub:4D798F6D-74BC-4FE8-BA46-79DAA314FE09. The online version of this work is archived and available from the following digital repositories: PeerJ, PubMed Central and CLOCKSS.

Systematic Paleontology

THERAPSIDA Broom, 1905	
THEROCEPHALIA Broom, 1903	
EUTHEROCEPHALIA Hopson & Barghusen, 1986	
WHAITSIOIDEA Tatarinov, 1974	

Composition—Theriognathus microps Owen, 1876; Ictidostoma hemburyi (Broom, 1911); Hofmeyria atavus Broom, 1935; Ictidochampsa platyceps Broom, 1948; Mirotenthes digitipes Attridge, 1956; Moschowhaitsia vjuschkovi Tatarinov, 1963; Viatkosuchus sumini Tatarinov, 1995; Microwhaitsia mendrezi gen. et sp. nov.; Ophidostoma tatarinovi gen. et sp. nov.

Table 1 List of tetrapod fossils collected from the two farms ‘Badshoek’ and ‘Good Luck’ by the Iziko South African Museum between 1996 and 2015.

The newly described therocephalian species are bold.

Specimen number	Identification	Stratigraphic level	
Good Luck (Matjiesfontein 412)	
SAM-PK-K11279	Dicynodon sp.	Steenkamp member (lower DaptoAZ)	
SAM-PK-K11189	Cistecephalus microrhinus	Oukloof/Steenkamp (uppermost CiAZ)	
SAM-PK-K11188	Mirotenthes digitipes	Oukloof/Steenkamp (uppermost CiAZ)	
SAM-PK-K8630	Pareiasaurus sp.	Oukloof member (upper CiAZ)	
SAM-PK-K8508	Diictodon sp.	Oukloof member (upper CiAZ)	
SAM-PK-K8509	Diictodon sp.	Oukloof member (upper CiAZ)	
SAM-PK-K8513	Diictodon sp.	Oukloof member (upper CiAZ)	
SAM-PK-K8303	Cistecephalus sp.	Oukloof member (upper CiAZ)	
SAM-PK-K8304	Cistecephalus sp.	Oukloof member (upper CiAZ)	
SAM-PK-K8510	Cistecephalus sp.	Oukloof member (upper CiAZ)	
SAM-PK-K8512	Cistecephalus sp.	Oukloof member (upper CiAZ)	
SAM-PK-K8629	Cistecephalus sp.	Oukloof member (upper CiAZ)	
SAM-PK-K11187	Cistecephalus sp.	Oukloof member (upper CiAZ)	
SAM-PK-K8307	Oudenodon sp.	Oukloof member (upper CiAZ)	
SAM-PK-K11280	Oudenodon sp.	Oukloof member (upper CiAZ)	
SAM-PK-K8507	Dicynodon sp.	Oukloof member (upper CiAZ)	
SAM-PK-K8516	Ophidostoma tatarinovi (type)	Oukloof member (upper CiAZ)	
SAM-PK-K8631	Eutherocephalia indet.	Oukloof member (upper CiAZ)	
SAM-PK-K8511	Procynosuchus delaharpeae	Oukloof member (upper CiAZ)	
SAM-PK-K11186	Procynosuchus delaharpeae	Oukloof member (upper CiAZ)	
Badshoek (De Hoop 117)	
SAM-PK-K10449	Pristerodon sp.	Oukloof member (CiAZ)	
SAM-PK-K11008	Pristerodon sp.	Hoedemaker member (upper TrAZ)	
SAM-PK-K11009	Pristerodon sp.	Hoedemaker member (upper TrAZ)	
SAM-PK-K11010	Pristerodon sp.	Hoedemaker member (upper TrAZ)	
SAM-PK-K11011	Diictodon sp.	Hoedemaker member (upper TrAZ)	
SAM-PK-K10987	Pristerodon sp.	Hoedemaker member (TrAZ)	
SAM-PK-K10985	Diictodon sp.	Hoedemaker member (TrAZ)	
SAM-PK-K10986	Emydops sp.	Hoedemaker member (TrAZ)	
SAM-PK-K11000	Tropidostoma sp.	Hoedemaker member (TrAZ)	
SAM-PK-K10983	Gorgonopsia indet.	Hoedemaker member (TrAZ)	
SAM-PK-K10984	Microwhaitsia mendrezi	Hoedemaker member (TrAZ)	
SAM-PK-K10990	Microwhaitsia mendrezi (type)	Hoedemaker member (TrAZ)	
Notes.

CiAZ Cistecephalus Assemblage Zone

DaptoAZ Daptocephalus Assemblage Zone

TrAZ Tropidostoma Assemblage Zone

Definition—The most inclusive clade that contains Theriognathus microps and Ictidostoma hemburyi, but not Ictidosuchus primaevus and Bauria cynops [stem-based].

Revised diagnosis—Small- to large-bodied therocephalians having a wide suborbital bar forming well-frontated orbits; ventromedially infolded maxilla with medially positioned tooth row so that much of the maxillary facial lamina is visible in ventral view; anterior border of orbit is located on anterior half of skull (conv. in cynodonts, Lycosuchus, some akidnognathids, and derived bauriamorphs); epipterygoid anteroposteriorly expanded; epipterygoid processus ascendens anterior edge orientation in lateral view is strongly anterodorsal (rather than posterodorsal or vertical) producing an anvil-shape; dentary tall and boomerang-shaped; and dentary ramus lateral groove weak (Ophidostoma) or absent (all others).

WHAITSIIDAE Haughton, 1918

Composition—Theriognathus microps Owen, 1876; Ictidochampsa platyceps Broom, 1948; Moschowhaitsia vjuschkovi Tatarinov, 1963; Viatkosuchus sumini Tatarinov, 1995; Microwhaitsia mendrezi gen. et sp. nov.

Definition—The most inclusive clade that contains Theriognathus microps and Viatkosuchus sumini, but not Ictidostoma hemburyi and Hofmeyria atavus [stem-based].

Revised diagnosis—Medium-to-large eutherocephalians with median frontonasal crest (also in Chthonosaurus and akidnognathids); suborbital vacuities reduced in size or absent; prefrontal and postorbital nearly contact, limiting contribution of frontal to dorsal border of orbit; epipterygoid extremely expanded anteroposteriorly (more so than hofmeyriids); epipterygoid posterior apophysis forms specialized “trigeminal notch”; pterygoid boss teeth usually absent (though present in Viatkosuchus).

MICROWHAITSIA MENDREZI gen. et sp. nov.	
(Figs. 2–6)	

Etymology—Micro (Greek, ‘small’); whaitsia (refers to whaitsiid affinities). Species epithet honors Christiane Mendrez-Carroll for her substantial contributions to the morphology and systematics of therocephalians.

Holotype—Iziko South African Museum (SAM) PK-K10990, partial skull with dentaries preserved in occlusion, missing most of the braincase, occiput, and postdentary bones (Figs. 2–4).

Figure 2 Holotypic skull of Microwhaitsia mendrezi gen. et sp. nov. (SAM-PK-K10990) in dorsal (A), ventral (B), and right lateral (C) views.

Figure 3 Interpretive line drawings of the holotypic skull of Microwhaitsia mendrezi gen. et sp. nov. (SAM-PK-K10990) in dorsal (A), ventral (B), and right lateral (C) views.

Abbreviations: d, dentary; C, upper canine; cr.ch, crista choanalis; f, frontal; f.l, lacrimal foramen; I5, fifth upper incisor; j, jugal; l, lacrimal; m, maxilla; n, nasal; p, parietal; pal, palatine; pC, upper precanine; PC5, fifth upper postcanine; po, postorbital; pm, premaxilla; prf, prefrontal; pt, pterygoid; sa, surangular; sm, septomaxilla; sp, splenial.

Figure 4 Stereopair images of the palate of the holotypic skull of Microwhaitsia mendrezi gen. et sp. nov. (SAM-PK-K10990) in left oblique ventral (A) and posteroventral (B) views.

Note the narrow contact between the left crista choanalis and vomer without sutural connection.

Referred specimen—SAM-PK-K10984, weathered snout preserving most of the left antorbital region, palate, and maxillary canine and postcanine alveoli (Figs. 5 and 6).

Figure 5 Referred specimen of Microwhaitsia mendrezi gen. et sp. nov. (SAM-PK-K10984) in dorsal (A), ventral (B), and left lateral (C) views.

Figure 6 Interpretive line drawings of referred specimen of Microwhaitsia mendrezi gen. et sp. nov. (SAM-PK-K10984) in dorsal (A), ventral (B), and left lateral (C) views.

Abbreviations: C, upper canine alveolus; cr.ch, crista choanalis; ect, ectopterygoid; f, frontal; j, jugal; l, lacrimal; m, maxilla; m.pal.f, maxillo-palatine foramen; n, nasal; pal, palatine; PC5, fifth upper postcanine alveolus; prf, prefrontal; pt, pterygoid; v, vomer; v.suborb, suborbital vacuity.

Locality and horizon—Both specimens were collected by RMHS in 2011 from Badshoek farm (De Hoop 117) near Beaufort West District, Western Cape Province, Republic of South Africa; upper Tropidostoma Assemblage Zone (Wuchiapingian stage), upper Permian Teekloof Formation (Hoedemaker member). Detailed locality information is available at Iziko South African Museum, Cape Town.

Diagnosis—Small-to-medium sized therocephalian with broad, robust skull; nasals strongly waisted at mid-length in dorsal view; thickened, pachyostotic frontal bone; thickened suborbital bar (nearly as deep as the orbit’s dorsoventral height); upper dental formula I5:pC1:C1:PC5; lower dental formula i4:c1:pc5.

General description

In general, the two specimens of Microwhaitsia show a robust snout, cheek, and dentition, and are similar to each other in overall size (estimated skull lengths ∼180 mm). The skull roofs are strongly sutured and craniofacial bones fairly thick for their small size. The holotypic skull–SAM-PK-K10990–is nearly complete with dentition and both lower dentaries intact in occlusion, but it is missing much of the occiput and intertemporal region, as well as the postdentary bones. The referred specimen–SAM-PK-K10984–is a weathered snout and left antorbital region, missing the premaxilla, and most of the right side of the skull. The dentition is also missing from the referred specimen. In both specimens, cranial sutures are easily distinguished and tend to be noticeably sinuous and interdigitating, with few straight sutures except along the midline. The braincase and occiput are not preserved in either specimen.

Skull roof

The premaxilla is a three-part element consisting of maxillary, vomerine, and dorsal (internarial) processes. The maxillary portion bears a short interdigitating suture with the maxilla that borders a small nervous foramen and, more dorsally, the septomaxillary foramen. Dorsally, the internarial process forms the medial septum of the naris and is fairly short, not reaching posteriorly beyond the level of the septomaxilla. The holotype preserves five premaxillary (incisor) alveoli. The septomaxilla is a rather large, solid element, but it is not well exposed outside of the naris. There is a modest dorsal process within the naris. The ventral footplate shares little overlap with the premaxilla externally, and the posterior facial process is fairly short dorsal and medial to the septomaxillary foramen.

The maxilla is the dominant element of the rostrum, having an exceptionally high facial lamina. The facial lamina is so high and broad and that very little of the nasal can be seen in lateral view (Figs. 2 and 3). It is highest anteriorly and, when viewed laterally, rapidly tapers just behind the anterior border of the orbit ventral to the jugal. The lateral surface of the rostrum is coarsely pitted by many nervous foramina in front of the antorbital depression. In ventral (palatal) view, the maxilla borders a fossa for the lower canine and expands behind this area forming a crista choanalis (Fig. 4). The crista is sufficiently expanded that it just contacts the vomer dorsally (Fig. 4B), separating the anterior (canine fossa) and posterior portions of the choana. However, there is no true maxillovomerine bridge as the connection is not sutural (unlike Theriognathus and Moschowhaitsia where the two elements are sutured). Attachment sites for maxilloturbinates on the medial side of the facial lamina could not be discerned.

The nasal is relatively long and narrow despite the broad, round profile of the snout. This is in part influenced by the large size of the maxilla facial laminae, which give the nasals a waisted appearance along their mid-length in dorsal view.

The lacrimal is a short, square element on the anterolateral margin of the orbit. It is much abbreviated due to the tall, broad facial lamina of the maxilla. Dorsally, it is bordered by the anterior tongue of the prefrontal, which contacts the maxilla and, thus, bars the lacrimal from contacting the nasal, as in most other therocephalians (except lycideopids). Just anterior to the orbit, near the contacts with the jugal and maxilla, there is a marked antorbital fossa which bears the lacrimal foramina (as in the therocephalian Hofmeyria and, convergently, burnetiamorph therapsids). One large foramen occupies a position on the anterior rim of the orbit, whereas the other smaller foramen is situated more ventrolaterally near the jugal-maxilla suture. The details of the nasolacrimal canal and its communication with the internal (medial) face of the maxilla could not be determined at present.

The prefrontal forms the anteromedial border of the orbit. It contacts the lacrimal ventrolaterally and the maxilla and nasal anteriorly. The posterior process does not contact the postorbital over the orbit, so that a small portion of the frontal contributes to the margin of the orbit. The prefrontal and postorbital nearly meet inside the orbit as in Theriognathus (Fig. 2C).

The frontal is very thick and pachyostotic (∼13 mm thick at midorbit level). In dorsal view, it is roughly diamond-shaped with stout lateral processes narrowly contacting the orbital margin. The element’s anterior extent just reaches the level of the anterior margin of the orbit where it strongly interdigitates with the nasal. There is a prominent median frontonasal crest on the midline as in other whaitsiids. The midline suture is comparatively straight except at the midorbit level where it becomes noticeably jagged.

The postorbital bounds the posteromedial margin of the orbit. The postorbital bar is robust and strongly sutures to the jugal. Most of the bar is formed by the broad postorbital process of the jugal. As in Theriognathus and other whaitsiids, the jugal is a substantial element, promoting the robust appearance of the skull despite its small size. However, the total depth of the suborbital bar is more so than in other whaitsiid genera, being 40–50% the height of the skull (nearly as tall as the orbit itself) (Figs. 2C and 3C). By contrast, hofmeyriids have comparatively thin suborbital bars (e.g., Hofmeyria, Mirotenthes).

Braincase and occiput

Most of the braincase and occiput are missing from the specimens. However, the bisected referred specimen clearly shows portions of the sphenethmoid preserved in place. The sphenethmoid complex has been described in a few therocephalians, including Ictidosuchoides (Crompton, 1955), Microgomphodon (Abdala et al., 2014a), Tetracynodon (Sigurdsen et al., 2012), and Theriognathus (Kemp, 1972a; Kemp, 1972b; Huttenlocker & Abdala, 2015). The anterodorsal portion of the sphenethmoid complex, the orbitosphenoid, resides just dorsal and posterior to the vomer, forming a broad but thin ascending lamina that would have made up the anteromedial wall of the orbit. As in Tetracynodon, there is a broadly convex notch along its posterior margin that may have accommodated the optic nerve (Sigurdsen et al., 2012: fig.2). Its dorsal most extent nearly contacts the ventral surface of the frontal, but is separated by a small fissure.

Palate, splanchnocranium, and suspensorium

The vomer is best viewed in palatal aspect in the holotype. It has a long interchoanal process that widens anteriorly as in other therocephalians and is widest at its anterior contact with the premaxilla. Although the crista choanalis just contacts the vomer anterodorsally, creating the appearance of a strongly vaulted palate, there is no true maxillovomerine bridge. There is a modest median crest that runs longitudinally along the vomer’s ventral margin, and a suture that separates the vomer into right and left halves spans the entire posterior region and much of the length of the interchoanal process (as in basal therocephalians and some hofmeyriid specimens).

The crista choanalis is short and smoothly rounded, rolling onto the palatine and terminating just anterior to the suborbital vacuity. As in other whaitsiids, there are at least two prominent ridges formed in this region: one that is confluent with the vomer-palatine suture emarginating the choana and a second that is more laterally positioned and confluent with the palatine-maxilla suture and alveolar margin (Figs. 5B and 6B). The maxilla-palatine foramen is positioned somewhat posteriorly, just medial to the first and second postcanines. The ectopterygoid is preserved on the left side of the skull in the referred specimen. It formed a narrow strut bracing between the palatine, maxilla, and jugal along the anterolateral margin of the suborbital vacuity. It is pierced ventrally by a small foramen. Little of the pterygoid could be discerned in either specimen. Likewise, the epipterygoid and the suspensorium (including the quadrate-quadratojugal complex) are not preserved in either specimen.

Mandible

Portions of the left and right lower jaws are preserved in the holotypic specimen, including the left and right dentaries and splenials, right coronoid, right prearticular, and right surangular. The dentary is a robust element with a thick mentum that was only loosely sutured at the symphysis. The overall shape is strongly bowed as in other whaitsioids. The lateral surface of the ramus is smooth, bearing no dentary groove or trough. The anterior surface of the mentum is pitted with fine vascular foramina. The coronoid process is very tall, terminating in the upper half of the orbit, and its terminal margin is slightly rounded. On the medial surface of the dentary, a long, flat splenial can be seen running anteriorly and meeting on the left and right sides just behind the dentary symphysis. Just posterior and dorsal to the splenial on the right side, a single coronoid and anterior portions of the prearticular are preserved, bounding the anterior and lower parts of the mandibular fenestra. The anterior half of the surangular is preserved on the medial surface of the dentary coronoid process and forms the upper border of the mandibular fenestra. The majority of the postdentary region is not preserved.

Dentition

The dentition is best preserved in the holotypic specimen, although maxillary alveoli are also present in the referred specimen. The holotype preserves five upper incisors, one upper precanine, one upper canine (a second root of a previously functional canine is also preserved on the left side of the skull immediately behind the newly functional canine), and five upper postcanines. The dentary preserves four lower incisors, a single lower canine, and five lower postcanines. As in most whaitsioids and early eutherocephalians in general, longitudinal fluting of the enamel was likely present on the incisors based on the fifth left upper incisor, which is the best preserved. However, most of the dentition is highly abraded, making the enamel texture difficult to examine. In general, the incisor crowns are subcircular in cross-section and lack serrations or mesio-distal carinae. The upper canine is fairly large, subcircular in cross-section, and also lacks serrations or carinae. Following a short diastema, the postcanines are preserved as simple, conical crowns that are subcircular in cross-section. These also lack any serrations or mesio-distal carinae. The surface enamel of the postcanines is too poorly preserved to discern any postcanine fluting.

WHAITSIOIDEA incertae sedis	
OPHIDOSTOMA TATARINOVI gen. et sp. nov.	
(Figs. 7 and 8)	

Etymology—Tatarinov’s ‘snake mouth.’ Ophido- (Greek, ‘snake’ or ‘serpent’); stoma (Greek, ‘mouth’). Genus name refers to the wide gape permitted by the loose mandibular symphysis and streptostylic jaw of whaitsioid therocephalians. Species epithet honors Leonid Tatarinov for his contributions to whaitsioid systematics and therocephalian functional anatomy.

Holotype—SAM-PK-K8516, weathered skull and articulated mandible missing portions of palate and braincase (Figs. 7 and 8).

Figure 7 Volumized HRXCT scans of the skull of Ophidostoma tatarinovi gen. et sp. nov. (SAM-PK-K8516) in dorsal (A), ventral (B), left lateral (C), and frontal (D) views.

Figure 8 Interpretive line drawings of the skull of Ophidostoma tatarinovi gen. et sp. nov. (SAM-PK-K8516) in dorsal (A), ventral (B), left lateral (C), and frontal (D) views.

Abbreviations: a, angular; d, dentary; ect, ectopterygoid; c, lower canine; C, upper canine; f, frontal; fen.m, mandibular fenestra; i, lower incisor; j, jugal; l, lacrimal; m, maxilla; n, nasal; p, parietal; part, prearticular; pbs, para-basisphenoid; PC6, sixth upper postcanine; po, postorbital; pm, premaxilla; prf, prefrontal; pt, pterygoid; q-qj, quadrate-quadratojugal complex; sa, surangular; sm, septomaxilla; sp, splenial; sq, squamosal; v.ipt, interpterygoid vacuity; v.suborb, suborbital vacuity.

Locality and Horizon—The specimen was collected by RMHS in 1996 on ‘Good Luck’ farm (Matjiesfontein 412) near Fraserburg District, Northern Cape Province, Republic of South Africa; upper Cistecephalus Assemblage Zone (Wuchiapingian stage), upper Permian Teekloof Formation (Oukloof member). Detailed locality information is available at Iziko South African Museum, Cape Town.

Diagnosis—Small therocephalian with short, broad skull (greatest skull breadth at occiput approximately equal to basal skull length); frontal length-to-width ratio approximately 1.0; pineal opening situated on raised boss; pterygoid boss teeth present; upper dental formula: I?:C1:PC6. Plesiomorphic in the retention of longitudinal striae/fluting on upper postcanines (primitive for eutherocephalians).

General description

The specimen consists of a small, poorly preserved skull and mandible (skull length from occipital condyle to tip of snout ∼60–65 mm), with the right side of the skull slightly crushed and the lower jaw laterally displaced. Its most recognizable features are the stout rostrum and the broad zygoma, producing a skull that is as wide as it is long.

The specimen was prepared using a pneumatic scribe so that most of the outer surface of the skull roof and mandible was exposed, and the ventral surface prepared to the palate. The skull was then scanned using high-resolution X-ray computed tomography (HRXCT) at the University of Utah Core Imaging Facility to reveal aspects of the palate, cranial sinuses, and braincase. It was scanned using an Inveon µ-CT scanner with scanner settings: 100 kV and 60 µA with a voxel size of 90 µm. Stacks were volumized and studied in OsiriX (Rosset, Spadola & Ratib, 2004). Unfortunately, the HRXCT data revealed that much of the palate and basicranium was ‘blown-out’ prior to fossilization, leaving only the skull roof, peripheral portions of the braincase and basicranium, and mandible.

The specimen has suffered from slight pre-burial distortion, including flaring of the right cheek and loss of the right zygoma, allowing ventrolateral displacement of the dentary ramus, although connection to the left dentary at the symphysis was maintained (Figs. 7D and 8D). The squamosal is also displaced slightly anteriorly within the temporal fossa on both sides. This distortion has led the authors to consider the possibility that the striking breadth of the skull relative to its short length was a result of postmortem crushing or other deformation. However, these proportions likely did not result from preservational artifact because large portions of the skull roof are preserved in articulation and some in situ cranial calvariae are clearly broad with their widths being subequal to or slightly greater than their lengths (e.g., frontal).

Skull roof

The maxilla is a very tall element with a broad facial lamina. The canine was deeply rooted, although it is mostly weathered and missing the entire crown. As in other whaitsioids, the facial lamina of the maxilla gently rolls onto the ventral (palatal) surface to meet the abbreviated alveolar margin (Figs. 7B and 8B). There are six closely packed postcanines situated along the alveolar margin on the right side of the skull. In palatal view, the choana can be seen to have been short and the crista choanalis also fairly short and rounded (rather than sharp medially). The medial surface of the facial lamina is smooth and shows no evidence for attachment of maxillary turbinates as in most other therocephalians (Huttenlocker, Sidor & Smith, 2011; Sigurdsen et al., 2012; but see Hillenius, 1994). Internally, there is a large, cavern-like anterior maxillary sinus (sensu Sigurdsen, 2006) as in other therocephalians that have been serially sectioned or CT scanned (Sigurdsen et al., 2012). The sinus opens anteriorly onto the medial surface of the maxilla. A posterior duct parallels the tooth row, leading into a smaller sinus (the posterior maxillary sinus; sensu Sigurdsen, 2006) that opens posteriorly into a shallow fossa on the medial surface of the maxilla just behind the tooth row.

The nasals are imperfectly preserved in SAM-PK-K8516, with most of the left side weathered away and the right badly fractured. Surface features and nasoturbinal ridges could not be discerned due to this damage. In life, the nasals would have been fairly short and broad as in the other skull roof elements.

The lacrimal is a square element that forms part of the anterior wall of the orbit and extends internally onto the palate. Two small nasolacrimal foramina located on the anterior margin of the orbit open into the nasolacrimal canal. The canal forms a narrow caliber duct that opens onto the medial surface of the facial lamina of the maxilla near the maxilla-lacrimal suture.

The prefrontal is a tall and broad element that forms most of the anteromedial margin of the orbit. It is fairly fractured on the right side of the skull and slightly weathered on the left so that its precise suture contacts are difficult to determine. However, given its breadth and the high facial lamina of the maxilla, it is likely that it contacted the maxilla anteriorly so that a nasolacrimal contact was not permissible as in most therocephalians.

The frontal forms a broad diamond that is as wide as it is long, a rare feature in a therocephalian. The interorbital breadth is therefore relatively wider than in other therocephalians. The dorsal surface is flat and smooth so that there is no median frontonasal crest. The frontal contributes strongly to the dorsomedial wall of the orbit, widely separating the prefrontal and postorbital. This contrasts with the condition in other whaitsioids where the latter elements contact in the upper wall of the orbit (hofmeyriids) or nearly contact along its dorsal margin (whaitsiids).

The postorbital is a lunate element forming most of the posterior border of the orbit. Its jugal process is long and robust, although its connection to the jugal is not preserved due to damage to the zygoma on both sides of the skull. More posteriorly, the postorbital broadly overlaps the parietal extending beyond the level of the pineal foramen.

The jugal is incompletely preserved, forming only a portion of the ventrolateral margin of the orbit and missing the postorbital and zygomatic processes. Based on the preserved portions, the suborbital bar was very slender as in Hofmeyria and Mirotenthes. The squamosal is disarticulated on both sides of the skull, being displaced anteromedially into the temporal fossa. It is tall and thin with a broadly concave anterior face that, when articulated, overlapped the supraoccipital, interparietal, and parietal. The sutural overlap with the parietal was extensive as in the postorbital.

The parietal is a tall element forming the dorsolateral walls of the braincase, which are vertical and parallel-sided as in other eutheriodonts. The parietal (sagittal) crest is short behind the pineal foramen. Internally, the enclosure forming the pineal canal is deep and the external opening large and situated on an elevated parietal boss. The interparietal could not be discerned with certainty due to crushing in this region and anterior displacement of the supraoccipital.

Braincase and occiput

Very little of the braincase and occiput are preserved in SAM-PK-K8516. The sphenethmoid complex could not be observed in the prepared specimen or in the CT scans. Small portions of the wedge-shaped supraoccipital are preserved on the left and right sides of the skull in close association with the parietal. Ventral to this, a tiny portion of the right exoccipital is preserved in contact with the basioccipital where the two would have bounded the foramen magnum. There is a strong basicranial girder formed between the ventromedian keel of the parabasisphenoid and the paired parasagittal processes of the pterygoid. The basal tubera are of moderate size (approximately 25% the width of the skull). The prootics and opisthotics are too poorly preserved to describe in detail.

Palate, splanchnocranium, and suspensorium

Much of the palate is unpreserved so that the relationships between the vomer, palatines and ectopterygoids cannot be described in detail. The maxillae bear a modest crista choanalis that border a fairly short choana. The crista is low and rounded, rather than sharp and distinct. Although the vomer is not preserved, a maxillovomerine bridge was likely to have been absent based on the lack of obvious medial (palatal) processes of the maxilla near the level of the canine or posterior to it. Portions of the pterygoid are preserved more posteriorly, showing well-developed transverse flanges that formed the posterior border of the suborbital vacuity. The flanges sweep medially and posteriorly where they formed paired parasagittal ridges bounding a wide interpterygoid vacuity. Pterygoid boss teeth are present, with three small, longitudinally arranged denticles present on the right and a large tooth preserved centrally on the left boss. The quadrate processes and the dorsally adjoining epipterygoid are not preserved. A massive element that rests within the quadrate recess of the squamosal on the left side of the skull is interpreted tentatively as the quadrate-quadratojugal complex.

Mandible

The mandible is preserved on the left and right sides of the skull. The left mandible is largely in place and held in occlusion, whereas the right is slightly ventrolaterally displaced and disarticulated. The dentary is long and strongly bowed, with the distinctive boomerang-shape that is common amongst whaitsioids. The coronoid process is broken on the left side, but is preserved on the right where it can be seen as an extremely tall element. This tall process would have terminated in the upper half of the temporal fenestra as in Hofmeyria and Mirotenthes. Its terminal margin is somewhat rounded. Along the ramus, there is a very modest lateral dentary groove (or trough), unlike in other whaitsioids that lack this feature entirely. Internally, there is a large nervous canal that runs the length of the dentary, dorsal to the Meckelian groove, and feeds the alveoli. Medially, portions of the prearticular are preserved in contact with the Meckelian groove in the left side. The dentary, prearticular, and angular border a fairly large mandibular fenestra as in other eutherocephalians. The angular bears a broad but thin reflected lamina with radiating ridges and a dorsal notch. The surangular borders the reflected lamina dorsomedially. The presence of an articular in the specimen could not be confirmed.

Dentition

Due to damage to the premaxilla, the upper incisors are not preserved in the specimen. The upper canines are large and deeply rooted, although their crowns have also weathered away almost completely. Six postcanines are present in the right maxilla, although the roots of only three are preserved on the left. The crowns are short, conical with circular cross-sections and lacking carinae, and bear fine longitudinal striae on their external surface as in some basal akidnognathids (i.e., Akidnognathus) and baurioids (i.e., Ictidosuchus). Just anterior to the position of the second postcanine, a replacement tooth can be seen emerging from the maxilla on the right side.

Most of the lower incisors are missing, although a single pair of incisors (presumably in the fourth position) is preserved on both the left and right dentary, each pressed against the mesial surfaces of the lower canines. Their surface texture is abraded such that the presence of enamel striae (as in the postcanines) cannot be confirmed. The tooth row is very short with most of the teeth (including the postcanines) restricted to the anterior third of the dentary. There are at least four (and likely more) lower postcanines packed tightly together behind the lower canine. The anteriormost postcanine in the right dentary preserves enamel striae.

Results & Discussion

Phylogenetic position of new whaitsioids

Whereas early cladistic investigations regarded hofmeyriids as relatively basal eutherocephalians (Hopson & Barghusen, 1986; Abdala, 2007), more recent studies have supported that they are most closely related to whaitsiids, such as Theriognathus (Huttenlocker & Abdala, 2015). Prior to the present study, the holotypic specimens of Microwhaitsia and Ophidostoma were incorporated into a preliminary phylogenetic analysis and were referenced only by their voucher numbers (Huttenlocker, 2014; Huttenlocker, Sidor & Angielczyk, 2015; Huttenlocker & Sidor, 2016). We assessed the phylogenetic positions of Microwhaitsia and Ophidostoma using a matrix of 136 craniodental and postcranial characters from 56 therapsid taxa (including two outgroups Biarmosuchus and Titanophoneus, plus 54 neotherapsid ingroup taxa) (see Appendices S1 and S2). The data were analyzed in PAUP*4.0b10 (Swofford, 1999) and MrBayes v. 3.2.6 (Ronquist, Van der Mark & Huelsenbeck, 2009). A heuristic search was performed in PAUP using maximum parsimony with a random addition sequence with 100 replicates and the tree-bisection-reconnection (TBR) branch-swapping algorithm. Characters were not ordered and all given equal weight. Multistate characters were interpreted as uncertainty. The Bayesian analysis was performed using the standard Mk model for morphological evolution (Lewis, 2001) with variable character rates. We ran the analysis for one million generations (with the first 25% removed as burn-in) and sampled the posterior distribution every 100 generations.

Parsimony analysis recovered 1,160 equally most parsimonious trees (MPTs) having a length of 381 steps (consistency index (CI) = 0.438; retention index = 0.789; rescaled CI = 0.346). The analysis found Hofmeyriidae, Whaitsiidae, and Baurioidea to represent monophyletic clades nested within Eutherocephalia. The high number of MPTs differed only in their volatile arrangements of basal scylacosaurid therocephalians, basal (ictidosuchid-grade) baurioids, and Triassic bauriids. Though the major (deep) branches of eutherocephalians are relatively stable, there were notable differences between parsimony and Bayesian-based approaches. In particular, the interrelationships of basal therocephalians are poorly resolved in the Bayesian topology, which fails to support the monophyly of Scylacosauridae. Moreover, the Bayesian topology yields a major eutherocephalian polytomy between Perplexisauridae, Chthonosauridae, Akidnognathidae, and the Whaitsioidea+Baurioidea clade. It also recovers the baurioid family Lycideopidae as the monophyletic sister-group to Bauriamorpha, while the parsimony trees recover Lycideopidae as a paraphyletic assemblage that includes Bauriamorpha as a subgroup. To account for these uncertainties, we present a conservative topology in Fig. 9 that illustrates the conflicting parsimony and Bayesian arrangements as polytomies.

Figure 9 Conservative phylogenetic relationships of the major clades of eutherocephalians, showing conflicting arrangements obtained from Bayesian and parsimony analyses as polytomies (tree length = 381; consistency index (CI) = 0.438; retention index = 0.789; res).

All of the MPTs retrieved a monophyletic Whaitsiidae composed of the Permian eutherocephalians Ictidochampsa (Broom, 1948), Viatkosuchus (Tatarinov, 1995), Moschowhaitsia (Tatarinov, 1963), and Theriognathus (Owen, 1876). Microwhaitsia is found to be nested within whaitsiids as the sister-taxon to Viatkosuchus + (Moschowhaitsia + Theriognathus). Among whaitsioids, characters shared between Microwhaitsia and other whaitsiids include: median frontonasal crest present; suborbital bar robust, deepened; postorbital bar moderately well built (extremely slender in hofmeyriids and baurioids); prefrontal and postorbital nearly contact on dorsomedial wall of orbit; upper postcanines reduced to five or fewer. Based on the character evidence, Viatkosuchus and Microwhaitsia would together represent the geologically oldest occurrences of whaitsiids in Laurasia and Gondwana, respectively (discussed further below).

Ophidostoma is found to be nested within Whaitsioidea, but the specimen does not demonstrate clear features that unite it with any given whaitsioid subgroup. Based on gross similarities to hofmeyriids, Huttenlocker, Sidor & Angielczyk (2015) labeled Ophidostoma as ‘Cistecephalus AZ hofmeyriid’ (Huttenlocker, Sidor & Angielczyk, 2015: fig. 6). However, the present analysis shows that Ophidostoma falls outside the clade that includes the closest common ancestor of Hofmeyriidae+Whaitsiidae, indicating that it is an aberrant whaitsioid and that its hofmeyriid-like gestalt represented plesiomorphies shared by all whaitsioids.

Whaitsioidea-Baurioidea dichotomy—The present phylogenetic hypothesis further implies a major dichotomy between two non-akidnognathid eutherocephalian radiations during the Permian: Whaitsioidea and Baurioidea. In general, the whaitsioids were characterized by small and large-bodied species (skull lengths ranging ∼6–35 cm) with a broad cranium having greater area for the jaw adductor musculature, a robust, boomerang-shaped mandible, and a tendency toward reduction of the postcanine teeth, whereas baurioids typically consisted of small to mid-sized species (skull lengths ranging ∼5–20 cm) with a low, slender rostrum and numerous postcanines. A close relationship between whaitsioids and baurioids was originally demonstrated by Huttenlocker (2009), who suggested that their common ancestry could be traced to no later than the middle-to-late Permian transition during Pristerognathus AZ times.

Figure 10 New specimen of the hofmeyriid Mirotenthes digitipes Attridge, 1956 (SAM-PKK11188) from the upper Cistecephalus Assemblage Zone of ‘Good Luck.’

Specimen shown in dorsal oblique view, showing the large temporal fenestra and broad, anvil-shaped epipterygoid (ept) processus ascendens.

Unlike Ivakhnenko’s (2011) usage of ‘Whaitsioidea’ which included lycosuchids and akidnognathids as subgroups—thus, making the name equivalent to Therocephalia—we define Whaitsioidea explicitly as the most inclusive clade that contains Theriognathus microps and Ictidostoma hemburyi, but not Ictidosuchus primaevus and Bauria cynops [stem-based]. As such, the group includes representative hofmeyriid and whaitsiid taxa (e.g., Hofmeyria, Theriognathus), but excludes akidnognathids and baurioids. Permian whaitsioids share a number of craniodental synapomorphies that distinguish them from the latter two groups, including: a wide suborbital bar forming well-frontated (i.e., forward-facing) orbits; ventromedially infolded maxilla facial lamina with concave alveolar margin (rather than convex as in some akidnognathids, or straight as in all others); anterior border of orbit located on anterior half of skull; anvil-shaped epipterygoid processus ascendens with anterior tilt (see below); broad contact between processus ascendens and parietal; dentary strongly bowed or boomerang-shaped; dentary ramus lateral groove/furrow weak (Ophidostoma) to absent. Some of these features have been discussed elsewhere (Huttenlocker, 2009; Huttenlocker, Sidor & Smith, 2011). Additional features of the braincase shared by whaitsioids may be elucidated with further computed tomographic research. The external morphology of the whaitsioid braincase also exhibits an anteriorly tilted epipterygoid with an anterodorsally oriented leading edge of the processus ascendens (Fig. 10) as demonstrated by a specimen of the hofmeyriid Mirotenthes (SAM-PK- K11188) recovered from coeval Cistecephalus AZ deposits at ‘Good Luck’ in association with Ophidostoma. This contrasts with the more upright or posteriorly leaning epipterygoid of baurioids (e.g., Sigurdsen et al., 2012: fig. 2) and other therocephalians (e.g., Huttenlocker, Sidor & Smith, 2011: fig. 4). The orientation of the processus ascendens has been incorporated as new character 136 in the present phylogenetic analysis.

Evolution of early eutherocephalian dental morphology and surface texture—The morphology and texture of the marginal dentition in therocephalians has only been discussed anecdotally, but may provide important information regarding the interrelationships of therocephalians, in addition to their diets (Huttenlocker, Sidor & Angielczyk, 2015). All eutherocephalians have lost the serrations on the canines and antecanine teeth, but few eutherocephalians may have retained smooth anterior and/or posterior carinae (e.g., chthonosaurids, some akidnognathids). In some Permian eutherocephalians, the incisors and precanines may bear longitudinal fluting (e.g., basal akidnognathids, whaitsioids, and ictidosuchid-grade baurioids) or large, flat facets (derived akidnognathids). Incisor fluting was suggested to be plesiomorphic for Eutherocephalia by Hopson & Barghusen (1986), although the postcanine enamel texture was not discussed in any detail. The condition of the incisors in Ophidostoma is unknown, but the presence of enamel striae on the postcanines strongly suggests that striated postcanine enamel was also plesiomorphic for the postcanines of early Eutherocephalia and its subclades, given its shared presence in Ophidostoma, Akidnognathus, and Ictidosuchus. Incidentally, enamel fluting has also been reported in the marginal dentition of the enigmatic scylacosaurian Scylacosuchus from Russia (Ivakhnenko, 2011). Contrary to the recent suggestion of Huttenlocker, Sidor & Angielczyk (2015: fig. 6) that basal akidnognathids and ictidosuchid-grade baurioids evolved postcanine striae independently, the new evidence from Ophidostoma suggests that a single evolutionary origin is more parsimonious, followed by multiple losses of enamel fluting/striae on the antecanine teeth and postcanines in derived Akidnognathidae, Whaitsioidea, and Baurioidea.

Diversification of therapsids during the end-guadalupian extinction

Non-marine vertebrate diversifications and extinctions are poorly understood during the middle-to-late Permian transition. Particularly, whereas most workers recognize marked extinctions of entire groups by the end of the Guadalupian (e.g., dinocephalians, lycosuchids, scylacosaurids), there is little consensus as to whether the proliferation of late Permian therapsid assemblages during Wuchiapingian times constitutes survival and opportunistic expansion of ecospace by pre-existing lineages (long-fuse) or a rapid radiation of new lineages (short-fuse) during the extinction’s aftermath (Fröbisch, 2008; Fröbisch, 2013; Lucas, 2017). For example, Lucas (2017) most recently characterized the extinction of dinocephalian faunas as an abrupt “global event” (p. 55), although systematic paleontological collecting in the middle Permian Tapinocephalus and Pristerognathus AZs suggests the turnover was more complex, and that the apparent severity of dinocephalian extinctions is exaggerated by oversplit dinocephalian taxonomy and poor temporal resolution (Rubidge et al., 2013; Day et al., 2015a; Day et al., 2015b). Nevertheless, others have identified shifts in overall extinction rates of some post-Guadalupian therapsid lineages—a potential driver of apparent (raw) diversity shifts in some Wuchiapingian lineages (Brocklehurst et al., 2015)—while others still have raised doubts about the quality of the fossil record and its ability to resolve the magnitude of turnover of middle-to-late Permian assemblages (Fröbisch, 2008; Fröbisch, 2013; Irmis, Whiteside & Kammerer, 2013).

We suggest that the apparent turnover of therocephalians can be characterized by a long-fuse model in which classic late Permian clades (e.g., whaitsioids, baurioids) originated concurrently with basal therocephalians of the middle Permian, but at lower abundances. This hypothesis is supported by renewed collecting efforts in the Teekloof Formation by one of us (RMHS), helping to further clarify therapsid ecological turnover during this time. Firstly, the new record of Microwhaitsia represents one of the oldest known Gondwanan whaitsiids, as other whaitsiid records from southern Africa are typically confined to the uppermost Cistecephalus and lower Daptocephalus AZs (Huttenlocker & Abdala, 2015). Microwhaitsia firmly establishes that whaitsiids already exhibited a Pangean-wide distribution by Tropidostoma AZ times (early Wuchiapingian), an observation that is consistent with the record of the closely allied Viatkosuchus from the Capitanian or Wuchiapingian-aged Kotelnich assemblage of Russia (Golubev, 2000; Benton et al., 2012) (Fig. 11). Secondly, long-term collecting efforts by one of us (RMHS) to elucidate the assemblages of the different members of the Teekloof Formation have produced additional whaitsioid and baurioid records (see Tables 2 and 3). Some of these—including a hofmeyriid (SAM-PK-K10525) (Fig. 12), an indeterminate ictidosuchid-grade baurioid (SAM-PK-K6886) and a second baurioid with possible affinities to Ictidosuchoides (SAM-PK-K11319)—were collected from the lower Poortjie Member or equivalent beds, which contains a Pristerognathus AZ fauna that corresponds to either the latest Capitanian or earliest Wuchiapingian global stages (Fig. 1).

Figure 11 Stratigraphically calibrated phylogeny of middle Permian through Triassic therocephalians showing calibration points for minimum divergence dates of major clades (A).

Light gray lines represent hypothetical phylogenetic branching, whereas black bars represent observed stratigraphic ranges (dashed ends indicate taxa having unknown upper or lower ranges). Graph (B) shows peak levels of eutherocephalian origination/extinction by the Wuchiapingian stage. Abbreviations: An, Antarctica; Ch, China; Chx, Changxingian; Ci AZ, Cistecephalus Assemblage Zone; Dapto AZ, Daptocephalus Assemblage Zone; Ind, Induan; Na, Namibia; Olen, Olenekian; Prist AZ, Pristerognathus Assemblage Zone; Roa, Roadian; Ru, Russia; SA, South Africa; TapinoAZ, Tapinocephalus Assemblage Zone; Tr AZ, Tropidostoma Assemblage Zone; Tz, Tanzania; Wor, Wordian; Za, Zambia.

Table 2 African Permo-Triassic therocephalians by Karoo assemblage zone or equivalent (updated from Abdala, Rubidge & Van den Heever, 2008 and Huttenlocker, 2013).

Permian	
Eodicynodon Assemblage Zone (Wordian) (2)	
Glanosuchus macrops	
Ictidosaurus angusticeps	
Tapinocephalus Assemblage Zone (Capitanian) (8)	
Alopecodon priscus	
Blattoidealestes gracilis*	
Crapartinella croucheri*	
Glanosuchus macrops	
Ictidosaurus angusticeps	
Lycosuchus vanderrieti	
Pardosuchus whaitsi	
Pristerognathus polyodon	
Scylacosaurus sclateri	
Simorhinella baini	
Pristerognathus Assemblage Zone (late Capitanian–Wuchiapingian) (5)	
Glanosuchus macrops	
Hofmeyriidae (cf. Hofmeyria)	
Ictidosuchidae (cf. Ictidosuchoides)	
Lycosuchus vanderrieti	
Pristerognathus polyodon	
Tropidostoma Assemblage Zone (Wuchiapingian) (6)	
Choerosaurus dejageri	
Hofmeyria atavus	
Ictidostoma hemburyi	
Ictidosuchoides longiceps	
Ictidosuchus primaevus	
Microwhaitsia mendrezi	
Cistecephalus Assemblage Zone (Wuchiapingian) (8)	
Mupashi migrator	
Euchambersia mirabilis	
Hofmeyria atavus	
Ichibengops munyamadziensis	
Ictidostoma hemburyi	
Ictidosuchoides longiceps	
Mirotenthes digitipes	
Ophidostoma tatarinovi	
Polycynodon elegans	
Theriognathus microps	
Unnamed akidnognathid (USNM PAL 412421)	
Daptocephalus Assemblage Zone (Wuchiapingian–Changxingian) (11)	
Akidnognathus parvus	
Cerdosuchoides brevidens	
Ictidochampsa platyceps	
Ictidosuchoides longiceps	
Ictidosuchops rubidgei	
Lycideops longiceps	
Mirotenthes digitipes	
Moschorhinus kitchingi	
Promoschorhynchus platyrhinus	
Tetracynodon tenuis	
Theriognathus microps	
Triassic	
Lystrosaurus Assemblage Zone (Induan–Olenekian) (7)	
Ericiolacerta parva	
Moschorhinus kitchingi	
Olivierosuchus parringtoni	
Promoschorhynchus cf. P. platyrhinus†	
Regisaurus jacobi	
Scaloposaurus constrictus	
Tetracynodon darti	
Cynognathus Assemblage Zone (Olenekian–Anisian) (2)	
Bauria cynops	
Microgomphodon oligocynus	
Notes.

* Taxa denoted by asterisk are considered invalid or based on non-diagnostic juvenile material.

† SAM-PK-K10014, originally identified as Ictidosuchoides (Smith & Botha, 2005; Botha & Smith, 2006; Abdala, Rubidge & Van den Heever, 2008; Huttenlocker, Sidor & Smith, 2011).

Table 3 Russian and Chinese Permo-Triassic therocephalians by assemblage zone or stage (updated from Abdala, Rubidge & Van den Heever, 2008, Ivakhnenko, 2011 and Huttenlocker, 2013).

Permian	
Wordian–Capitanian (Ulemosaurus Assemblage Zone, Isheevo fauna or equivalent) (2)	
Perplexisaurus(?) lepusculus*	
Porosteognathus efremovi	
Late Capitanian–Wuchiapingian? (Deltavjatia Assemblage Zone, Kotelnich fauna or equivalent) (6)	
Karenites ornamentatus	
Kotelcephalon viatkensis	
Perplexisaurus (=Chlynovia) foveatus	
Scalopodon tenuisfrons	
Scalopodontes kotelnichi	
Viatkosuchus sumini	
Wuchiapingian (Proelginia Assemblage Zone, Ilynskoe fauna or equivalent) (1)	
Scylacosuchus orenburgensis	
Wuchiapingian-Changxingian? (Scutosaurus Assemblage Zone, Sokolki fauna or equivalent) (2)	
Annatherapsidus petri	
Chthonosaurus velocidens	
Changxingian (Archosaurus Assemblage Zone, Vyazniki fauna or equivalent) (6)	
Dalongkoua fuae†	
Hexacynodon purlinensis*	
Malasaurus germanus*	
Moschowhaitsia vjuschkovi	
Purlovia maxima	
Whaitsiidae indet.	
Triassic	
Induan–Olenekian (Vetlugian stage or equivalent) (5)	
Hazhenia concava	
Scalopognathus multituberculatus*	
Silphedosuchus orenburgensis	
Urumchia lii	
Yikezhaogia megafenestrala	
Anisian (Eryosuchus Assemblage Zone, Donguzian fauna or equivalent) (7)	
Antecosuchus ochevi	
Dongusaurus schepetovi†	
Nothogomphodon danilovi	
Nothogomphodon sanjiaoensis	
Ordosiodon lincheyuensis	
Ordosiodon youngi	
Traversodontoides wangwuensis	
Notes.

* Taxa denoted by asterisk are considered invalid or based on non-diagnostic material.

† Precise age uncertain. Regarded as Permo-Triassic by Liu & Abdala, 2017.

Figure 12 Representative hofmeyriid from the late Capitanian or earliest Wuchiapingian of the Karoo Basin, South Africa, compared to other known specimens of Hofmeyria.

SAM-PK-K10525 (A), Hofmeyriidae from the Pristerognathus Assemblage Zone of Lombardskraal, Beaufort West district, Western Cape Province. Specimen shows short, high rostrum, prefrontal-postorbital contact in orbit, anteriorly expanded epipterygoid, and conical, non-serrated/non-carinated maxillary teeth. BP/1/4404 (BP, former Bernard Price Institute, now Evolutionary Studies Institute, Johannesburg) (B), Hofmeyria cf. H. atavus from the Cistecephalus Assemblage Zone of Matjiesfontein (Highlands), Victoria West district, Northern Cape Province. BP/1/1399 (C), Hofmeyria cf. H. atavus from the Cistecephalus Assemblage Zone of Driehoeksfontein, Murraysburg district, Western Cape Province. Numbers 14–136 are characters listed in the phylogenetic analysis (see online Supplemental Information 1) followed by the derived state in parentheses corresponding to hofmeyriids or other early whaitsioids. Abbreviations: AZ, Assemblage Zone; PC, postcanine position #.

Origination, extinction, and diversification rate shifts in therocephalians have been summarized elsewhere in the context of the Permo-Triassic mass extinction (Huttenlocker, Sidor & Smith, 2011; Huttenlocker, 2013; Huttenlocker, 2014). For example, Huttenlocker (2014) failed to identify evidence of diversification rate shifts in small-bodied therocephalian lineages near the Permo-Triassic boundary, but noted a significant shift associated with the earlier divergence of the Eutherocephalia clade. This was attributed to either increasing origination rates of eutherocephalians or to sampling bias in the Tropidostoma and Cistecephalus AZs where therocephalian fossils are more abundant (Smith, Rubidge & Van der Walt, 2012). More recently, Brocklehurst et al. (2015) showed that uneven origination and extinction rates (particularly elevated extinction) could be a driver of diversification rate shifts in Permo-Triassic tetrapods. Notably, both origination and extinction rates of therocephalians rose steadily into the late Permian (Fig. 11), despite the long lineage durations sustained by a few representative taxa during the Wuchiapingian (e.g., Ictidosuchoides, some hofmeyriids). In this sense, origination/extinction rates indicate that therocephalians (particularly eutherocephalians) were, on the whole, resilient to the effects of the extinction, although this was dependent upon high rates of turnover and replacement by individual genera. Unlike some baurioid lineages, whaitsioids became wholly extinct prior to the Permo-Triassic boundary as therocephalian origination rates began to markedly decrease. The reasons for this differential extinction remain unclear, but have been linked to differences in life history strategies evident in the two groups (Huttenlocker & Botha-Brink, 2013; Huttenlocker & Botha-Brink, 2014; Botha-Brink et al., 2016). Consequently, future collecting of precious middle-to-late Permian specimens will be crucial to further resolve the evolutionary dynamics of therocephalians spanning the end-Guadalupian and Permo-Triassic mass extinctions.

Conclusions

The new records of Microwhaitsia and Ophidostoma from the Teekloof Formation shed light on the early evolution of eutherocephalians during the middle-to-late Permian transition. Phylogenetic analysis recovers both of the new taxa within Whaitsioidea, with Microwhaitsia as an early whaitsiid and Ophidostoma as an aberrant whaitsioid outside the hofmeyriid+whaitsiid clade. Consequently, Microwhaitsia represents the oldest whaitsiid from Gondwana and, along with additional early hofmeyriid and baurioid records, underscores the early dichotomy between whaitsiid-line and baurioid-line therocephalians. Moreover, the disjunct geographic occurrences of Microwhaitsia and Viatkosuchus suggest that whaitsiids already exhibited a cosmopolitan distribution by the early Wuchiapingian. During the end-Guadalupian, the extinction of basal lycosuchids and scylacosaurids was offset by increasing origination/extinction rates of eutherocephalians, which flourished into Wuchiapingian times. As a part of this radiation, whaitsioids represent a previously underappreciated but successful clade of late Permian eutherocephalians, but they did not survive the ecological impacts of the Permo-Triassic mass extinction. Future collecting will provide added resolution on the middle-to-late Permian transition, and will further clarify the dynamic replacement of basal therocephalians (lycosuchids, scylacosaurids) by eutherocephalians and other early-diverging therapsid predators (gorgonopsians, cynodonts).

Supplemental Information

Supplemental Information 1 Supplementary Appendices

Click here for additional data file.

File S1 Supplementary Data File

Click here for additional data file.

For collections access, we thank S Kaal and the Iziko South African Museum, Cape Town, and B Rubidge and B. Zipfel of the Evolutionary Studies Institute (former Bernard Price Institute, BP), Johannesburg. For comments on SAM-PK-K8516 and help in the field, we thank S Modesto. CT scans of SAM-PK-K8516 were produced by the University of Utah Core Imaging Facility, Salt Lake City.

Additional Information and Declarations

Competing Interests

Author Contributions

Data Availability

New Species Registration

The authors declare there are no competing interests.

Adam K. Huttenlocker conceived and designed the experiments, performed the experiments, analyzed the data, contributed reagents/materials/analysis tools, wrote the paper, prepared figures and/or tables, reviewed drafts of the paper.

Roger M.H. Smith conceived and designed the experiments, performed the experiments, analyzed the data, contributed reagents/materials/analysis tools, wrote the paper, reviewed drafts of the paper.

The following information was supplied regarding data availability:

Specimens described or figured here are reposited in the Iziko South African Museum, Cape Town, and at the Evolutionary Studies Institute, University of the Witwatersrand, Johannesburg.

The following information was supplied regarding the registration of a newly described species:

Publication LSID: urn:lsid:zoobank.org:pub:4D798F6D-74BC-4FE8-BA46-79DAA314FE09

Genus name (Microwhaitsia): urn:lsid:zoobank.org:act:0777041D-6950-41AE-B3BA-9A43E2526086

Genus name (Ophidostoma): urn:lsid:zoobank.org:act:81945DA4-5206-42D7-86E3-B4A9CFC892B1.

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
