# Peer review of "New whaitsioids (Therapsida: Therocephalia) from the Teekloof Formation of South Africa and therocephalian diversity during the end-Guadalupian extinction"

_PeerJ, doi:10.7717/peerj.3868_

## Round 0.1 · original submission · Minor Revisions

Dear Dr Huttenlocker

Your manuscript #19251, entitled "New whaitsioids (Therapsida: Therocephalia) from the Teekloof Formation of South Africa and therocephalian diversity during the end-Guadalupian extinction " which you submitted to PeerJ has been reviewed by two reviewers and the editor.

Both reviewers consider that your contribution is suitable for publication and should be accepted after minor revision. In this context, the reviewers have pointed out several changes concerning misspellings, missing references, and rewording, among others. Reviewer #2 has marked them in the annotated manuscript.

So, I am requesting that you address the suggestions mentioned above and resubmit your manuscript to PeerJ.

Looking forward to receiving your revision.

Sincerely, Claudia Marsicano

·

Basic reporting

This paper describes two new whaitsioids, and the description is informative. I agree on the identification of new taxa, but I think it is better to compare them with relative taxon to show their autopomorphies and other diagnostic characters. This part can add after the phylogentic analysis.

Figure 5 A the light is too bright
B The position of this one seems not match the direction of light

ONLINE SUPPLEMENTARY INFORMATION is nearly identical to 2015 JVP paper, you need adjust it for this paper. Really, only the codings of new taxa are needed here.

Experimental design

For the diversity study, my opinion is similar to that of Frobisch and Irmis. You can do it and you need think about the number of specimens as well. Your main point is the earlier appearance of Hofmeyria and other species. It is better you provide a photo of SAM-PK-K10525 and other specimens (L635-L637), because they are your key evidence.

Validity of the findings

no comment

Additional comments

You said the age of Pristerognathus Assemblage Zone is (late Capitanian–Wuchiapingian) (Table 2, L917), but your figure 1 and figure 11 it is only in Capitanian.
Also, you state Hofmeyria atavus in Pristerognathus Assemblage Zone, why do you not show it on Fig 11?

Table 3 Dalongkou fuae We showed the age is latest Permian in Fig. 1

L130 miss a ‘.’
L172, 173 References: Broom 1905, 1903
L266, L 274 repeat ‘bars the lacrimal from contacting the nasal’
L278 Fig. 3C?
L289 bar is more so than in other?
L301 dorsalmost
L432 anterior face that (should) overlap the It is disarticulated!
L437 enclosure forming the pineal duct??
L582 Ancestral state reconstructions of early eutherocephalian dentition
The title is too big for the following content, you only state morphology and texture, change it

L588 precanine maxillary teeth?
Figure captions: remember change the taxonomical names in italic
L603 For this kind of study, my opinion is similar to Irmis. You can do it and you need think about the number of specimens as well.
L692 Micro

·

Basic reporting

No comment

Experimental design

No comment

Validity of the findings

No comment

Additional comments

This manuscript names and describes for the first time two previously mentioned but unnamed whaitsioid therocephalian taxa, which were in the past years collected from the upper Permian Teekloof Formation of the South African Karoo Basin. The manuscript is well written and requires only very minor editorial corrects (see track changes in doc file). The content is clearly articulated and well structured. The illustrations are sufficient and properly referenced throughout the manuscript. Beyond the description, the authors discuss the new specimens' relationships within Therocephalia and their impact on the general phylogeny of the clade. Moreover, the Discussion includes a brief treatment of ancestral therocephalian dentition (as new anatomical information from the new material provides new insights) as well as a short review of therapsid and specifically therocephalian diversification across the end-Guadalupian Extinction. All of these aspects are well integrated and provide new insights into therocephalian evolution. The only concern that I have with regards to this contribution concerns the fact that the authors assume a mostly undistorted nature of the holotype and only specimen of the second new genus, Ophidostoma. The authors mention that the type skull is slightly crushed but on the other hand disregard the possibility that the overall unique skull proportions of being as wide as long is a result of distortion rather than genuine. I don't generally doubt the validity of this taxon or the overall anatomy as described, but the images and drawings clearly indicate that the skull is massively distorted. This should be made more clear in the text and overall proportions should be taken with caution. Other than that and the few suggestions for changes (see doc file), I strongly recommend publication of this manuscript in PeerJ.

---

## Round 0.2 · Minor Revisions

Dear Adam,

Your recently submitted revision of your Ms (#2017:07:19251) and your rebuttal letter has been evaluated.

Accordingly, I am sending to you an annotated new version with some minor comments on it that I consider will certainly improved the final version of your Ms.

Looking forward for the new version of your Ms.

Sincerely, Claudia Marsicano

---

## Round 0.3 · accepted · Accept

Dear Adam

It is a pleasure to accept your Ms, co-authored with Roger Smith, which you submitted to PeerJ.

Thank you for your fine contribution. We look forward to your future contributions to the Journal.

cheers,

Claudia Marsicano